# Analysing the evolution of aerospace ecosystem development

**Luna A. Jose, Jr.** [ID][1,2]*, **Alexandra Brintrup**[1]☯, **Konstantinos Salonitis**[2]☯

**1** Department of Engineering, Institute for Manufacturing, University of Cambridge, Cambridge, Cambridgeshire, United Kingdom, **2** Manufacturing Department, School of Aerospace, Transport and Manufacturing, Cranfield University, Cranfield, Bedfordshire, United Kingdom

☯ These authors contributed equally to this work.
* j.luna@cranfield.ac.uk, jjl57@cam.ac.uk

**Data Availability Statement:** The data for our analysis include exports figures from 1992 to 2016 obtained from the United Nations (UN), using Standard International Trade Classification (SITC) revision 3. The data acquisition was conducted

## Abstract

Aerospace manufacturing industry is predicted to continue growing. Rising demand is triggering the current global aerospace ecosystem to evolve and adapt to challenges never faced before. New players into the aerospace manufacturing industry and the development of new ecosystems are evidencing its evolution. Understanding how the aerospace ecosystem has evolved is thus essential to prepare optimal conditions to nurture its growth. Recent studies have successfully combined economics and network science methods to map, analyse and predict the evolution of industrial ecosystems. In comparison to previous studies which apply network science-based methodologies to macro-economic research, this paper uses these methods to analyse the evolution of a particular industrial ecosystem, namely the aerospace sector. In particular, we develop bipartite country-product networks based on trade data over 25 years, to identify patterns and similarities in the evolution of developed aerospace manufacturing countries ecosystems. The analysis is elaborated at a macroscopic (network) and microscopic (nodes) levels. Motivated by studies in ecological networks, we use nestedness analysis to find patterns depicting the distribution and evolution of exported products across ecosystems. Our analysis reveals that developed ecosystems tend to become more analogous, as countries lean towards having a revealed comparative advantage (RCA) in the same group of products. Countries also tend to become more nested in their aerospace product space as they start developing a higher RCA. It is revealed that although countries develop an advantage on unique products, they also tend to increase competition with each other. Further analysis shows that manufactured products have a stronger correlation to an aerospace ecosystem than primary products; and in particular, the automotive sector shows the highest correlation with positive aerospace sector evolution. Competition between countries with well-developed aerospace ecosystems tends to centre on automotive parts, general industrial machinery, power generating machinery and equipment, and chemical materials and products.

during May – July 2018, from the UN Comtrade database available online at https://comtrade.un.org/.

**Funding:** JJLA. Scholar: 328001, scholarship: 440378. This research was supported by CONACYT (Consejo Nacional de Ciencia y Tecnologia – the Mexican National Council for Science and Technology - https://www.conacyt.gob.mx/) and CONCYTEQ (Consejo de Ciencia y Tecnología del Estado de Queretaro). The funders had no role in study design, data collection and analysis, decision to publish, or preparation of the manuscript.

**Competing interests:** The authors have declared that no competing interests exist.

## Introduction

In 2018, results from all commercial airlines worldwide published by the International Air Transport Association (IATA) exhibited that the passenger traffic grew 7.4%. The increase is still dominated by North America and Europe (12.8%), followed by the Asia-Pacific region (9.5%), Latin America (7%), Africa (6.1%) and Middle East (5%) [1]. Furthermore, in the following years, the aerospace industry is predicted for continued future growth [2]. Over the next twenty years, passenger traffic figures are projected to double up. The most substantial market demand is expected to swing to the Asia-Pacific region, overtaking America and Europe's position [2–4].

As of 2016, the aerospace manufacturing industry has been dominated by the Airbus and Boeing duopoly (for commercial aeroplanes), and by GE Aviation, Pratt & Whitney, Rolls-Royce, Safran Aircraft Engines (for engines, which signify nearly 30% of a full aeroplane cost). Around 80% of world exports in aerospace have been generated by the United States (USA)– 29%, France (FRA)– 22%, Germany (DEU)– 12%, the United Kingdom (UK)– 9%, Canada (CAN)– 4%, and Italy (IT)– 2% [5].

Even though the market demand is growing, the aerospace ecosystem has not been able to react as needed. Aeroplane manufacturers have consistently reported insufficient capacity to cope with production requirements, reaching in 2018 a record high commercial aeroplane backlog of more than 14,000 units [6]. Key inhibitors like considerably high investment necessary for the development of new players and for the relocation of production facilities have limited its evolution [5]. In spite of the constraints, there are countries and companies that have overcome previously mentioned barriers. For instance, the introduction of new players to the aerospace manufacturing industry, and the development of new ecosystems by relocating manufacturing facilities in emerging aerospace manufacturing countries, like Mexico. Indeed, the current global aerospace ecosystem is tied to keep evolving and adapting to attend challenges never faced before. Understanding how the aerospace ecosystem has evolved is thus essential to prepare optimal conditions to nurture its growth.

Scientists have analysed and tried to explain the behaviour of industrial systems by applying an ecosystem approach, analogously from biological systems. The term ecosystem has been applied in different contexts since its first appearance. It was first introduced in 1935 by a British ecologist named A.G. Tansley, where he defined an ecosystem as a biological system located in a particular physical environment integrated by interactive and interdependent organisms [7]. Many years later, in 1993, James F. Moore, an American business strategist, adopted for the first time this biological approach to business theory by introducing the concept of a business ecosystem. Moore defined a business ecosystem as a sustainable economic community integrated by evolving and adapting self-organised organisations and individuals that interact with each other to survive [8]. In this paper, the term ecosystem is used with the aim of taking a holistic approach by embracing all the exported goods that nurture the portfolio of a specific country.

Understanding industrial ecosystems using network science has recently gained interest within researchers, as it is considered as a powerful approach to represent, analyse and predict its evolution [9]. Along the same line, this research applies network science aiming to help the development of emerging aerospace ecosystems, by understanding how developed aerospace ecosystems have evolved.

To perform this research, first, we collect historical international trade data from 1992 to 2016. Then, we compute the revealed comparative advantage (RCA) on aerospace products. We calculate the RCA for the rest of the product portfolio of selected countries, and then we identify their correlation with the aerospace exports. Then, we develop bipartite country-

product networks and identify patterns and similarities in the evolution of developed aerospace manufacturing countries ecosystems. Among our main findings is that developed ecosystems tend to become more analogous, as countries lean towards having an RCA>1 in the same group of products. Furthermore, our analysis also helps to identify which particular industries have nourished the growth of the aerospace ecosystems over a twenty-five years period.

## Literature review

The beginning of the XXI century has ignited the application of network science as a powerful approach for representing and analysing industrial ecosystems [10–14]. Network Science is defined as the "study of collection, management, analysis, interpretation and presentation of relational data" [12]. Some studies have successfully applied network science to develop economic theories and predict evolution, based on developing country-product networks. One of the first attempts was in the XIX century, when [15] claimed for the first time that countries benefitted mainly by specialising on products on which they have demonstrated a comparative advantage. More recently, [16] claimed that developing countries tend to have high product diversification, while developed countries tend to specialise in niche products. However, a few years later, [17–19] used historical international trade data to predict countries' product diversification, and reported that developed countries are highly diversified and have numerous products with an RCA>1. They also highlighted that developing economies have historically developed a comparative advantage only on products that are also exported by countries with high product diversification. [18,20] introduced an alternative methodology to Hidalgo and Hausmann for analysing countries' export flows and product diversification. Based on biased Markov chains, they ranked countries in a conceptually consistent approach and revealed a non-linear interaction among the catalogue diversification and the universality of products of a country. More recently, Hartmann et al. (2017) used multivariate regression analysis on the country-product networks to demonstrate that levels of income equality in a country are related to the complexity of their exported products.

Along the same line, there is a subset of studies that have used network science for a particular business ecosystem. For instance, [21] used trade data of the garment industry to analyse its disassembly process and to test a model of declining networks. [22] used a database of around 40,000 firms of the automotive industry to analyse the topology of Toyota's supply chain. They claim that the tier structure of Toyota's supply chain creates a complexly woven network, rather than a pyramidal structure as previously theorised. [23] proposed a framework to analyse the topological robustness of manufacturing industry and validated it using a dataset from the automotive industry. They evidenced that network science can be applied to study structural interdependencies of large-scale data. [24] combined agent-based model, discrete event modelling and network science to simulate the evolution of the consumption-driving supply chain system of the automotive industry in China. [25] analysed the structure of the aerospace industry using Airbus' supply chain consisting of 544 companies with more than 1,600 interactions between them. Here, authors demonstrated that the large-scale dataset analysed is a supply network formed by communities connected by interconnected hub firms. They also evidenced that network science can be applied to identify crucial firms within a network, and that is useful mainly to propagating information. [26] analysed the network evolution of the European aerospace ecosystem using data from the European Framework Programmes and on Airbus suppliers. They investigated the spatial structure of the European aerospace R&D collaboration network, the topological structure, the individual elements of the network, and an evaluation of the Airbus invention and production networks. Among their findings is that

these type of networks are formed by well-connected hubs, and that the regional hub structure is emulated in topology of the European aerospace R&D collaboration network. Also, they claim that only successful firms are the ones capable to form a vast amount of ties. [27] also analysed the evolution of the aerospace ecosystem by using a dataset consisting of firm linkages within 52 aerospace clusters in North America and Europe. To analyse the evolution and dynamics of the topological structure, they divided the dataset into three periods: 2002–2005, 2006–2009 and 2010–2014. They evidenced that the topology of networks have evolved across the different periods, and that clusters have increasingly specialised in value chain stages over time.

In tandem, motivated by studies in ecology, scientists have analysed nestedness patterns in networks across a variety of fields. The concept of nestedness originated in ecology and was introduced to describe patterns in two types of bipartite networks: mutualistic interaction patterns between species-species networks, and distribution patterns across species-habitat networks. A bipartite network is characterised for being partitioned into two classes without ties within classes [14]. Mutualistic interaction patterns are found in networks where two different species interact and beneficiate reciprocally. The interaction between insects and plants, when insects feed and pollinate from plants at the same time, are examples of mutualistic networks [28,29]. The pattern found within these networks is that most common interactions occur between generalist insects and plants, and between specialists with generalists, but not between specialists with specialists. Here, generalist insects refer to those feeding from multiple plants and generalist plants to those having many pollinators/feeders, while specialists are insects feeding from a small number of plants and plants having few pollinators/feeders. The second type of networks was individually conceived in biogeography by [30–32] to describe distribution patterns of species across isolated habitats [11,33,34]. Examples include the distribution of species within islands. Here, the distribution pattern found is that generalist islands congregate a vast number of species, while specialist islands host proper subsets of species existing in generalist islands. The pattern also suggests that rare species are most likely to exist in generalist islands rather than in specialist ones.

After being unveiled in ecology, nestedness patterns have been discovered across networks of different nature. For instance, patterns found in inter-organisational networks. [35] developed a model to reproduce the structure of manufacturer-contractor interactions, in which they found that these type of networks depict a similar pattern than the mutualistic interaction patterns between species-species networks. Nestedness patterns have also been found in supply chain networks by [36]. Here, authors analysed a large dataset of the automotive industry, particularly from the Toyota Motor Company and the Ford Motor Group, to demonstrate that supply networks of this industry depict nestedness patterns. They showed that generalist companies are the only ones producing specialist products and that specialists companies compete practically utterly in the generalist products market. Another study of nestedness patterns in supply chains is presented in [37]. Here, they analysed the supplier-product distribution and supplier-manufacturer relations in the global automotive industry. They claim that specialist suppliers produce proper subsets of what generalist suppliers produce, and that specialist products are only produced by generalist suppliers. Also, they found that specialist manufacturers procure from generalist suppliers, and specialist suppliers normally supply to generalist manufacturers.

Another type of networks in which nestedness patterns have been found is in trade networks. For instance, [34] developed country-product networks using trade data from 1985 to 2009, connecting 114 countries to 772 different products. Here, they developed a model to predict the evolution of business ecosystems by analysing the dynamics of nestedness, positing that nestedness arises when an industrial ecosystem has a core set of interactions attached to

the rest of the community. [18] used trading data of around 200 countries and 1200 products to introduce a new metric to assess the competitiveness of a country and the complexity of its product portfolio. [38] developed a dynamic network formation model to examine the topological structure and nestedness in real-world networks. They empirically tested their model using two different types of networks, the banking network and trade network between countries. [39] analysed the evolution of country-product networks, using trade data from 1995 to 2010, aiming at the identification of early symptoms of the 2007–2008 financial crisis. They evidenced that the structure of the network started to experience significant changes since 2003, and suggested that the most critical early signs are found in the macro-sectors evaluated on developing countries.

Although the analysis of networks using network science approach has been growing in the last years, it could be alleged that this approach is still in its infancy compared to other fields [9]. Moreover, while most studies that use economics and network science-based methodologies have thus far focussed on the macro-economic space, few studies have combined and applied such methodologies to understand the evolution of particular ecosystems. In this research, this gap will be approached to some extent by developing an analytical approach for a particular industry, namely the aerospace ecosystem.

## Methodological approach

Data were collected from 1992 to 2016 obtained from the United Nations (UN) Comtrade database. Using an RCA analysis, two groups of countries were selected. One group of countries that have been consistently among the top aerospace exporters, and another group of countries that have shown significant improvement on aerospace exports (we complemented the study by identifying all the other products with an RCA>1 for each selected country). Aiming at the identification of patterns across different periods, the 25 years data was divided into periods with an equal amount of years. Thus, five periods of five years were identified to formulate the analysis. For each period and country, a correlation analysis was performed to identify the strength of the statistical relationship between the RCA value on aerospace products and the RCA values of all the other exported products.

A total of ten bipartite, unweighted and undirected networks were produced (five networks per group of countries). Each graph is defined as $G = (N,E)$ comprising:

- $N = X \cup Y$ set of nodes, where $X$ are countries and $Y$ are products with $RCA>1$.

- $E \in X \cap Y$ set of edges, where a connection is made only when a specific product $Y$ has an RCA>1 at that country $X$.

In addition, the colour of $E$ depicts the Pearson correlation coefficient ($\rho$). Red edges indicate $\rho \geq 0.5$ and black edges all the others.

Finally, to identify evolution patterns we examined and compared the networks' topology using network and node-level's metrics, including a nestedness analysis. The detailed procedure is described in the *Network Analysis* section.

## Data collection

The data for our analysis include exports figures from 1992 to 2016 obtained from the United Nations (UN), using Standard International Trade Classification (SITC) revision 3. The data acquisition was conducted during May–July 2018, from the UN Comtrade database available online at https://comtrade.un.org/. The source data used for the analysis was selected as it is claimed to be the most complete trade database available worldwide [40]. This source has also been commonly used by researches to develop economic theories [19,20,41–43].

There are two commodities' classifications available: Harmonised System (HS) and SITC. The first one is mainly used by countries to collect their trade statistics. The latter one, which is the one selected for this analysis, is maintained by the United Nations (UN) and recommended for analytical purposes [44,45]. Within the SITC nomenclature, there were four revisions available at the time when data was collected: revision 1 containing data from 1962, revision 2 containing data from 1976, revision 3 with data from 1986 and revision 4 with data from 2007. Revision 3 was chosen as it is the latest classification with more than twenty years of historical data. Older revisions were not considered as there is no available data for some countries.

SITC nomenclature is grouped in 5 different levels to classify products according to their origin, where each level is represented by one digit. The most detailed level is the five-digit classification. However, one of the limitations described by the UN statistic division is that countries do not necessarily report data for each level and each year [46]. Thus, it was concluded that the two-digit classification was the most appropriate given the lack of data for more detailed levels.

After analysing all commodity codes and levels under revision 3, it was noted that there is not a commodity code that comprises all aerospace manufacturing products. For instance, commodity code '792 –Aircraft, associated equipment' seems to include all aerospace manufacturing products. However, it does not include products such as '7131 –Aircraft piston engines' or '82111 –Seats of a kind used for aircraft'. Consequently, a new code was proposed to encapsulate all aerospace products: 'code A: aerospace and associated equipment' (Table 1). Duplicates were avoided by subtracting modified codes from its upper levels.

To facilitate the analysis, groups of commodities were used as presented in Table 2, based on the statistical office of the European Union (Eurostat) classification [47]. Data is classified into primary and manufactured products. Primary products are those traded as found in nature, whereas manufactured products are goods processed from primary products. Subsequently, we proposed groups of products based on their industrial origin.

**Data assumptions and limitations.** The UN Comtrade database has more than 3.3 billion records with detailed exports and imports of around 200 countries and more than 6,000 different products [40]. According to [46], the following limitations should be considered when using SITC nomenclature for analytical purposes. First, all the data available is shared with the UN Statistics Division by the statistical authorities of each country, where countries do not necessarily provide data for every year and nomenclature level. Consequently, the UN does not estimate any missing data that was not reported by a country. To address this issue, where we considered necessary, we obtained the missing values by following three possible paths. The first way was by consulting trade databases available for each country. If no information was obtained, we estimated its value by using the exports' share average of the six nearest years of data available. In the case when a few data were available (less than 20 years available), we decided to exclude the commodity from our dataset. The commodity codes excluded are: '91 –

**Table 1. Code A: Aerospace and associated equipment.**

| Code | Description |
|---|---|
| 6253 | Tyres, pneumatic, new, of a kind used on aircraft |
| 7131 | Aircraft piston engines |
| 714 | Engines, motors non-electric |
| 792 | Aircraft, associated equipment |
| 82111 | Seats of a kind used for aircraft |
| 88571 | Instrument panel clocks and clocks of a similar type, for vehicles, aircrafts |

Mail not classed by kind', '93 –Special transactions not classified', '96 –Coin non-gold and non-current' and '97 –Gold, non-monetary and excluding ores'. In regards to the products included within the exports figures, SITC revision 3 considers entrepot or bonded warehouse trade, re-exports, trade-in bunkers and stores with foreign ships and aircraft, but it does not include goods passing through the country for purposes of transport only. In regards to the defence sector, there is not a unique commodity code used to classify products from this origin. To clarify this issue, we raised the concern to the UN statistics division. The answer obtained is the following: "Military goods can be part of UN Comtrade if they are reported as such by countries; however, for some countries, data for this type of commodity trade is confidential. In the latter case, the commodity may be identified at the chapter level but not at the 5-digit level, or it may just be lumped under 93 –Special transactions not classified". Therefore,

**Table 2. Group of commodities proposed by the authors.**

| Type | Group | Code | Product |
|---|---|---|---|
| **Manufactured products** | **Aerospace Products** | A | Aerospace and associated equipment |
| | **Automotive Products** | 78 | Road vehicles (automotive products) |
| | **Chemicals** | 51 | Organic chemicals |
| | | 52 | Inorganic chemicals |
| | | 53 | Dyeing, tanning and colouring material |
| | | 55 | Perfume, cleaning and preparations |
| | | 56 | Fertilisers, manufactured |
| | | 57 | Plastics in primary forms |
| | | 58 | Plastics in non-primary forms |
| | | 59 | Chemical materials and products |
| | **Machinery** | 71 | Power generating machinery and equipment |
| | | 72 | Machinery for specialised industries |
| | | 73 | Metalworking machinery |
| | | 74 | General industrial machinery |
| | | 75 | Office machines and adapted machines |
| | | 76 | Telecommunications and sound recording equipment |
| | | 77 | Electric machinery and parts |
| | **Metals** | 67 | Iron and steel |
| | | 68 | Non-ferrous metals |
| | | 69 | Manufactures of metals |
| | **Miscellaneous Products** | 62 | Rubber manufactures |
| | | 63 | Wood and cork manufactures |
| | | 64 | Paper, paperboard and articles thereof |
| | | 66 | Non-metallic mineral manufactures |
| | | 81 | Prefabricated buildings, sanitary, lighting and fixtures |
| | | 82 | Furniture and parts thereof |
| | | 83 | Travel goods, handbags and similar containers |
| | | 87 | Instruments and apparatus |
| | | 88 | Photographic equipment, optical goods |
| | | 89 | Miscellaneous manufactured articles |
| | **Pharmaceutical Products** | 54 | Medicinal and pharmaceutical products |
| | **Textiles and Clothing** | 61 | Leather, dressed fur |
| | | 65 | Textile yarn, fabrics, made-up articles |
| | | 84 | Articles of apparel and clothing accessories |
| | | 85 | Footwear |

*(Continued)*

**Table 2.** (Continued)

| Type | Group | Code | Product |
|---|---|---|---|
| **Primary Products** | **Transport Equipment** | 79 | Other transport equipment |
| | **Agricultural Products** | 00 | Live animals |
| | | 01 | Meat and meat preparations |
| | | 02 | Dairy products and birds' eggs |
| | | 03 | Fish and fish preparations |
| | | 04 | Cereals and cereal preparations |
| | | 05 | Vegetables and fruit |
| | | 06 | Sugars, sugar preparations and honey |
| | | 07 | Coffee, tea, cocoa, spices |
| | | 08 | Feeding stuff for animals |
| | | 09 | Miscellaneous edible products and preparations |
| | | 11 | Beverages |
| | | 12 | Tobacco and tobacco manufactures |
| | | 21 | Hides, skins, fur skins, raw |
| | | 22 | Oilseeds, oleaginous fruits |
| | | 23 | Crude rubber (incl. synthetic) |
| | | 24 | Cork and wood |
| | | 26 | Textile fibres and their wastes |
| | | 29 | Crude animal, vegetable materials |
| | | 41 | Animal oils and fats |
| | | 42 | Fixed vegetable fats and oils |
| | | 43 | Processed animal or vegetable oils |
| | **Energy** | 32 | Coal, coke and briquettes |
| | | 33 | Petroleum and products |
| | | 34 | Gas, natural and manufactured |
| | | 35 | Electric current |
| | **Non-Agricultural Raw materials** | 25 | Pulp and waste paper |
| | | 27 | Crude fertilizers and crude minerals |
| | | 28 | Metalliferous ores and metal scrap |

defence sector's products are considered under our analysis only if countries report this data to the UN.

For China, we combined the individual administrative regions (SAR) into one single value. Meaning that exports' figures of China considered in our analysis constitute values from China, plus Hong Kong and Macao.

## Revealed comparative advantage

Understanding that raw exports figures do not necessarily provide accurate evidence on the capability of a country to export product, we searched for a metric suitable for our study. The RCA was chosen as it has been widely used in academic and economic analyses [48]. RCA is based on comparing the exports of a specific country with the exports of the rest of the world (1). An RCA>1 depicts that a country has a relative advantage of exporting a specific product; the higher RCA value, the higher advantage.

$$RCA = \frac{\frac{Country's\ Exports\ of\ Specific\ Product}{Country's\ Total\ Exports}}{\frac{World\ Exports\ of\ Specific\ Product}{Total\ World\ Exports}} \tag{1}$$

Two groups of countries are needed for our analysis: one group that has been consistently among the top on aerospace products, and another group of countries that have improved their exports capability on aerospace products by moving from RCA<1, calculated using code A.

Results evidence that the countries with the most developed ecosystems (group one–G1) are FRA, the UK and the USA. For group two (G2), we selected Brazil (BRA), CAN and DEU as they evolved from an emerging aerospace ecosystem to an ecosystem with an RCA>1. Fig 1A illustrates the total amount of exports of the selected countries, while Fig 1B shows the aerospace exports only; both figures evidence countries of group one with higher numbers. Fig 1C depicts the RCA evolution. Here, the improvement of countries of group 2 is evidenced when crossing the RCA trigger in 1998.

The next step is the identification of other products that have consistently demonstrated an RCA>1 in both groups of countries.

## Correlation analysis

Pearson Correlation analysis was used to identify the strength of the statistical relationship between aerospace products and other RCA>1 goods exported by each country. Only positive

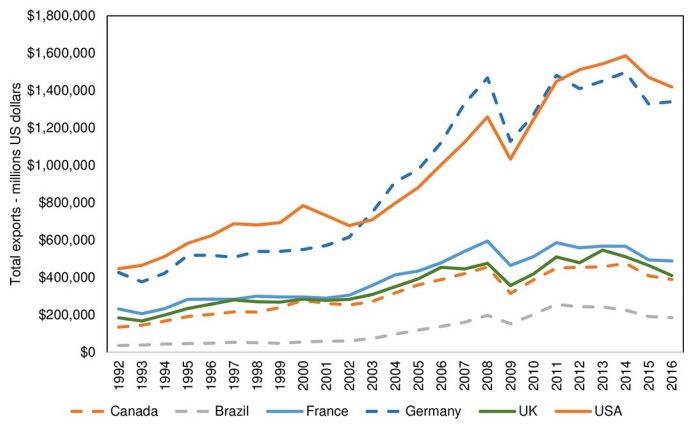

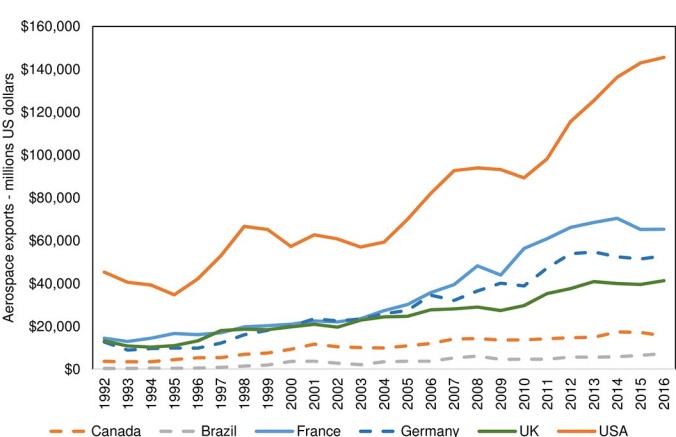

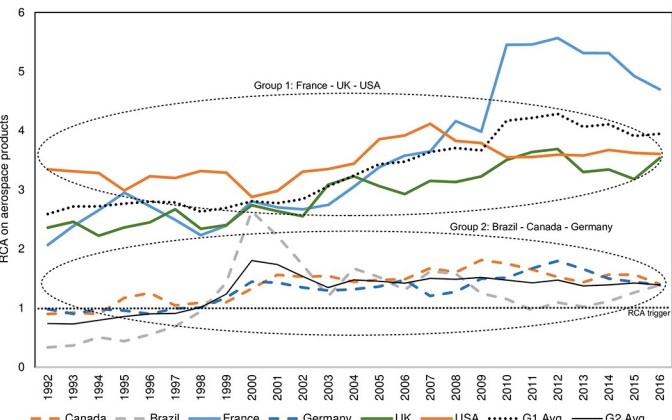

**Fig 1. a.** Total exports. Million Dollars (USD) of all products exported by the selected countries. **b.** Aerospace exports. Million Dollars (USD) of aerospace products exports. **c.** Revealed comparative advantage of aerospace products. Evolution of RCA on aerospace products using code A for calculations (RCA>1 depicts that the country has an RCA on exporting aerospace products).

**Table 3. Example of correlation calculations between RCA of code 'A' and '78' for FRA.**

|  | 78 | A |  | 78 | A |  | 78 | A |  | 78 | A |  | 78 | A |
|---|---|---|---|---|---|---|---|---|---|---|---|---|---|---|
| 1992 | 1.20 | 2.07 | 1997 | 1.26 | 2.49 | 2002 | 1.47 | 2.67 | 2007 | 1.36 | 3.65 | 2012 | 1.16 | 5.57 |
| 1993 | 1.17 | 2.39 | 1998 | 1.26 | 2.23 | 2003 | 1.49 | 2.74 | 2008 | 1.29 | 4.16 | 2013 | 1.10 | 5.31 |
| 1994 | 1.21 | 2.65 | 1999 | 1.30 | 2.39 | 2004 | 1.60 | 3.05 | 2009 | 1.34 | 3.98 | 2014 | 1.09 | 5.31 |
| 1995 | 1.23 | 2.95 | 2000 | 1.43 | 2.81 | 2005 | 1.53 | 3.38 | 2010 | 1.27 | 5.45 | 2015 | 1.07 | 4.92 |
| 1996 | 1.25 | 2.72 | 2001 | 1.44 | 2.70 | 2006 | 1.44 | 3.58 | 2011 | 1.28 | 5.46 | 2016 | 1.06 | 4.70 |
| Correlation | 0.68 | | | 0.89 | | | - 0.10 | | | - 0.86 | | | 0.93 | |

correlations were considered ($\rho \geq 0.5$), as we are interested in those relationships where aerospace exports rise by increasing the exports of any other product. The data was divided into the following periods: 1992–1996, 1997–2001, 2002–2006, 2007–2011 and 2012–2016. An example of the RCA values of code '78 –Road Vehicles' and 'A–Aerospace and Associated equipment' for FRA is given in Table 3.

Graphs for each period and group of countries are created in an undirected, unweighted and bipartite form. Fig 2 illustrates an example of the networks developed. Two classes of nodes exist: countries and goods. Only goods with an RCA>1 are represented by nodes in each graph. The colour and label of the nodes are related to the group of commodities presented in Table 2; blue nodes are manufactured products, and green nodes are primary products; FRA and CAN are represented with grey nodes; the UK and DEU with red nodes; and BRA and the USA with black nodes. Edges are used to connect the products with an RCA>1 to each country of study, which means that commodity's nodes are connected with the country's nodes only where RCA>1. Edges are also used to represent a correlation between exporting aerospace products at each country and any other commodity. Red edges depict a positive correlation ($\rho \geq 0.5$), and grey edges depict a correlation below this value. The networks developed for group 1 are presented in Fig 3A–3E and for group 2 in Fig 3F–3J.

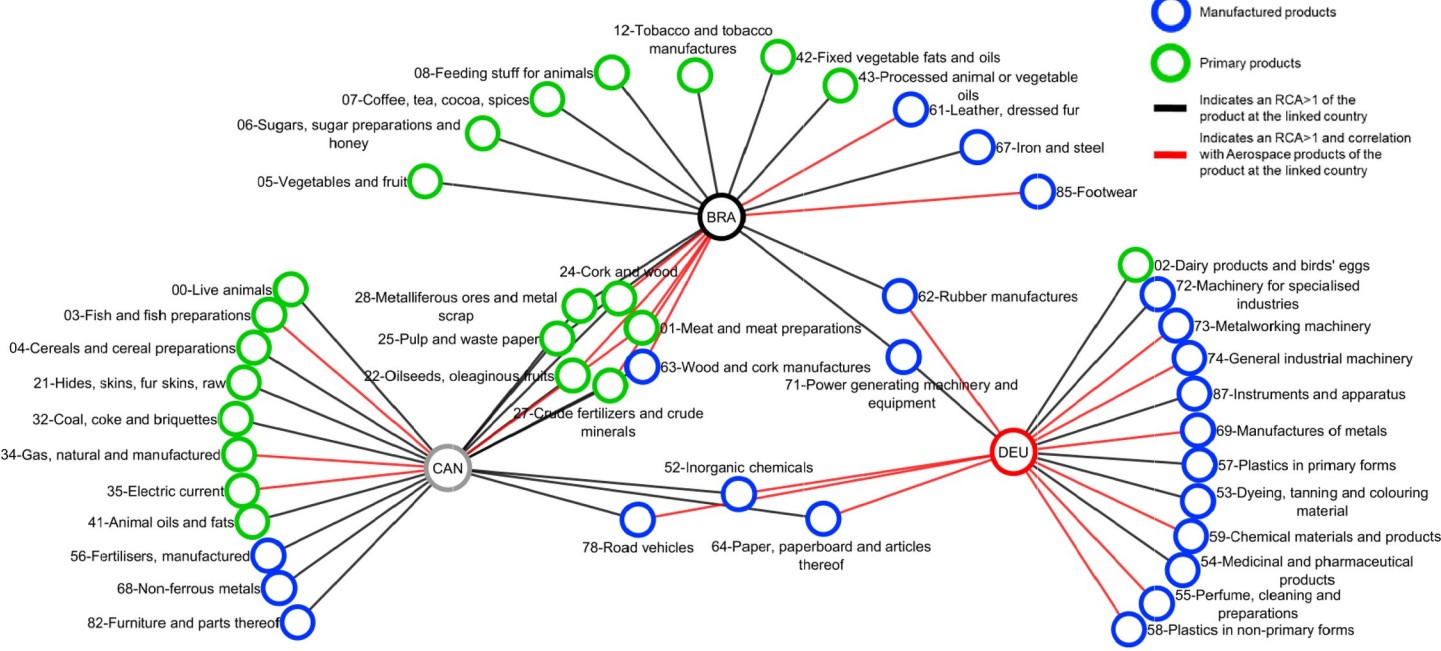

**Fig 2. Country-product network structure.**

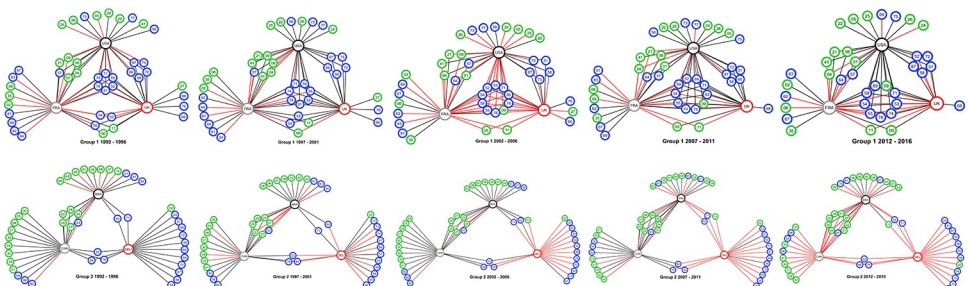

**Fig 3. Bipartite country-product network. a.** Group 1 1992–1996. **b.** Group 1 1997–2001.**c.** Group 1 2002–2006. **d.** Group 1 2007–2011. **e.** Group 1 2012–2016. **f.** Group 2 1992–1996. **g.** Group 2 1997–2001. **h.** Group 2 2002–2006. **i.** Group 2 2007–2011. **j.** Group 2 2012–2016.

## Network analysis

Networks' topology can be analysed from macroscopic and microscopic levels. The macroscopic level refers to the properties that can be observed at a network scale, while the microscopic level analyses properties that typify the particular position of an individual node in a network [10,12]. The following metrics were selected at the macroscopic level: *centralisation*, *density*, *matrix temperature*, *NODF*; at the microscopic level: *degree centrality*.

**Macroscopic level.** *Centralisation* measures how the connectedness of a network is distributed around particular nodes. *Density* measures the relationship between actual and potential connections within a network. While a high value of network centralisation reveals that connections are centralised in fewer nodes, a low value reflects that the power is more equally distributed. In regards to the density of a network, the highest value is when all nodes are connected with all others [10,49].

Fig 4 illustrates the evolution of the network centralisation and network density for both groups. Here, it is evidenced that as the aerospace ecosystem evolves, country-product networks tend to increase their cohesiveness and to distribute the power across fewer nodes. This is aligned with the RCA evolution, where both groups improved their aerospace ecosystem capability. As illustrated, across all periods of study, the group with a less developed aerospace ecosystem has lower values of centralisation and density than the developed ones. Group 2 developed a minor increase across the analysis, with an overall increase lower than 10% in both metrics. In contrast, the group of developed aerospace ecosystems experienced an increase higher than 20% in both measures, evidencing that centrality and density of the country-product networks increases as their ecosystem improves.

**Nestedness analysis.** Nestedness was introduced in ecology to describe patterns of two types of bipartite networks: species-species and species-habitat networks. The first one raises as a result of an interaction between two different species, in which both of them benefit from the interaction. The interaction between insects and plants, pollinators/feeders-plants, are examples of mutualistic networks [28,29]. The second type is used to describe the distribution patterns of species across isolated habitats. The study of the geographical distribution of species within islands are examples of these networks [30–32]. Inspired by previous studies, scientists have emulated the nestedness approach from ecology to other types of networks, such as social networks, inter-organisational networks, supply chain networks and country-product trade networks.

Aligned with previous studies, this research analyses nestedness patterns across the evolution of country-product trade networks of the aerospace ecosystem. Particularly, this study emulates the mutualistic networks approach to identify patterns on the distribution of products with an RCA>1 among the evolution of aerospace ecosystems.

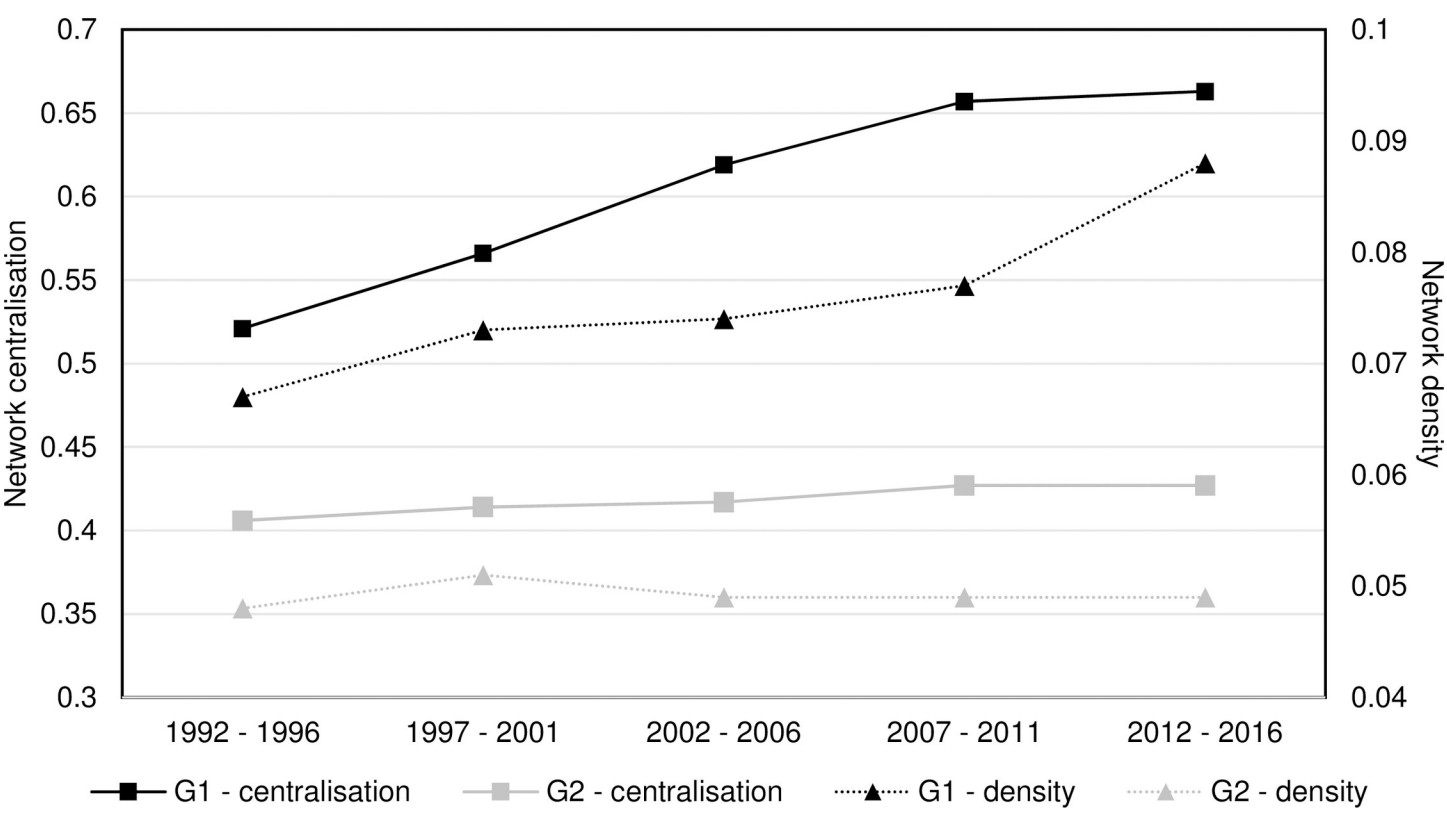

**Fig 4. Networks' centralisation and network density.**

Nested patterns are analysed by using packed adjacency matrix representing relations in the bipartite networks. A packed matrix is formed when columns and rows are sorted in decreasing order according to the marginal sums, starting in the upper rows and left-hand columns [33]. In our analysis, the unpacked matrices are organised by products' code and countries' RCA average (RCA average for each period); in the packed matrices, country-product nodes with the highest degree are grouped in the top left corner. Fig 5A–5E illustrate the unpacked matrices for group 1 and Fig 6A–6E show the packed matrices, which are formed following the rules previously mentioned. The unpacked and packed matrices for group 2 are illustrated in Fig 7A–7E and Fig 8A–8E, respectively.

Most of the nestedness metrics are based on measuring either the gaps or the columns versus rows overlapping of the adjacency matrix. For instance, *matrix temperature (T)*, and *Brualdi and Sanderson (BR)* also named discrepancy (amount of absences) measures are gap based metrics, while *nested overlap and decreasing fill (NODF)* is an overlap counting metric. *T* is intrinsic to the spreading of gaps inside the matrix. A lower *T* depicts more order inside the matrix, meaning that presences are concentrated in the upper left corner; it represents the average residual from the *isocline of perfect nestedness (IPN)* [50]. The range is from 0 to 100, where 0 represents a perfectly nested matrix. In terms of countries-exports ecosystems, a lower temperature means the most popular products have a majority distribution in most popular countries. *BR* metric counts the number of absences or presences that must be modified to generate perfect nestedness [51]. The fewer number of discrepancies, the more nestedness. *NODF* metric computes whether the occurrences of unpopular products within most popular countries, and whether depauperate country-product groupings represent subsets of the mighty ones [33]. The range is from 0 to 100, where 100 indicates perfect nestedness.

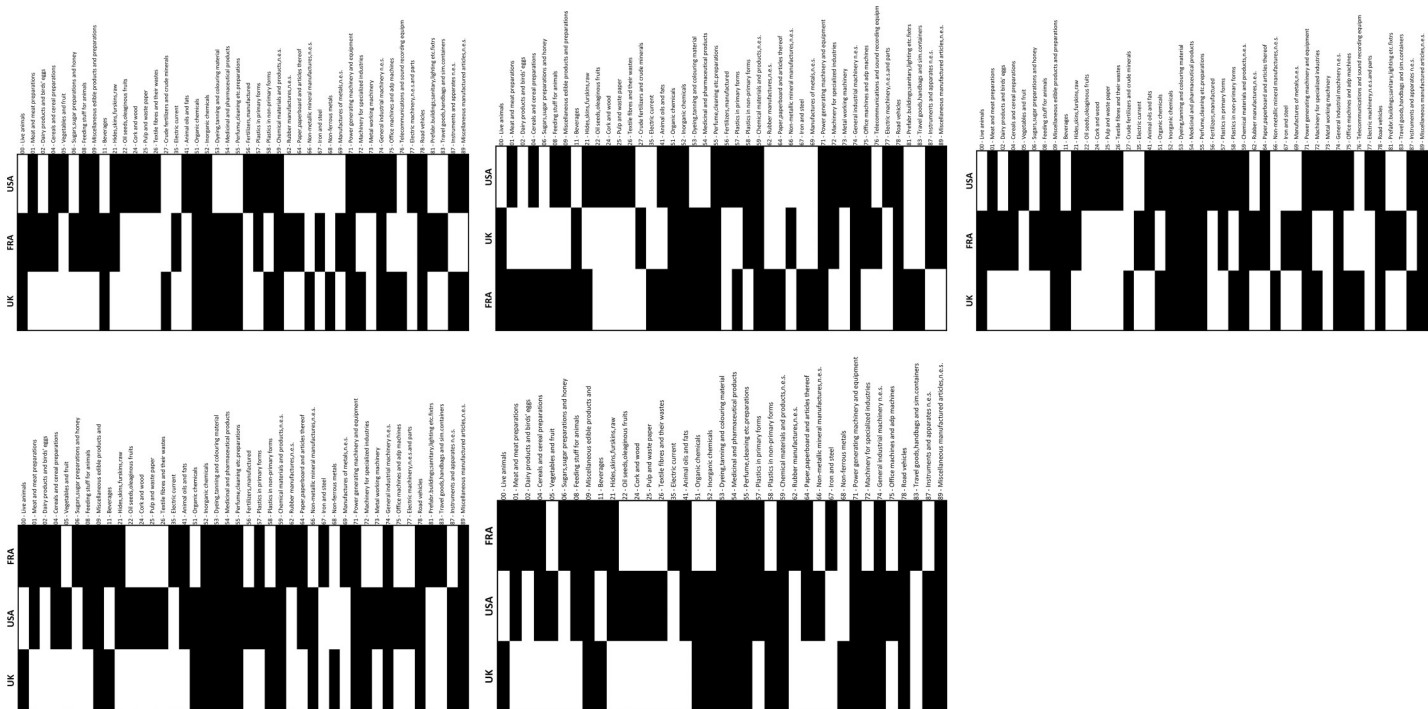

**Fig 5. Evolution of unpacked matrices. a.** Group 1: FRA, the UK and the USA, 1992–1996. **b.** Group 1: FRA, the UK and the USA, 1997–2001. **c.** Group 1: FRA, the UK and the USA, 2002–2006. **d.** Group 1: FRA, the UK and the USA, 2007–2011. **e.** Group 1: FRA, the UK and the USA, 2012–2016.

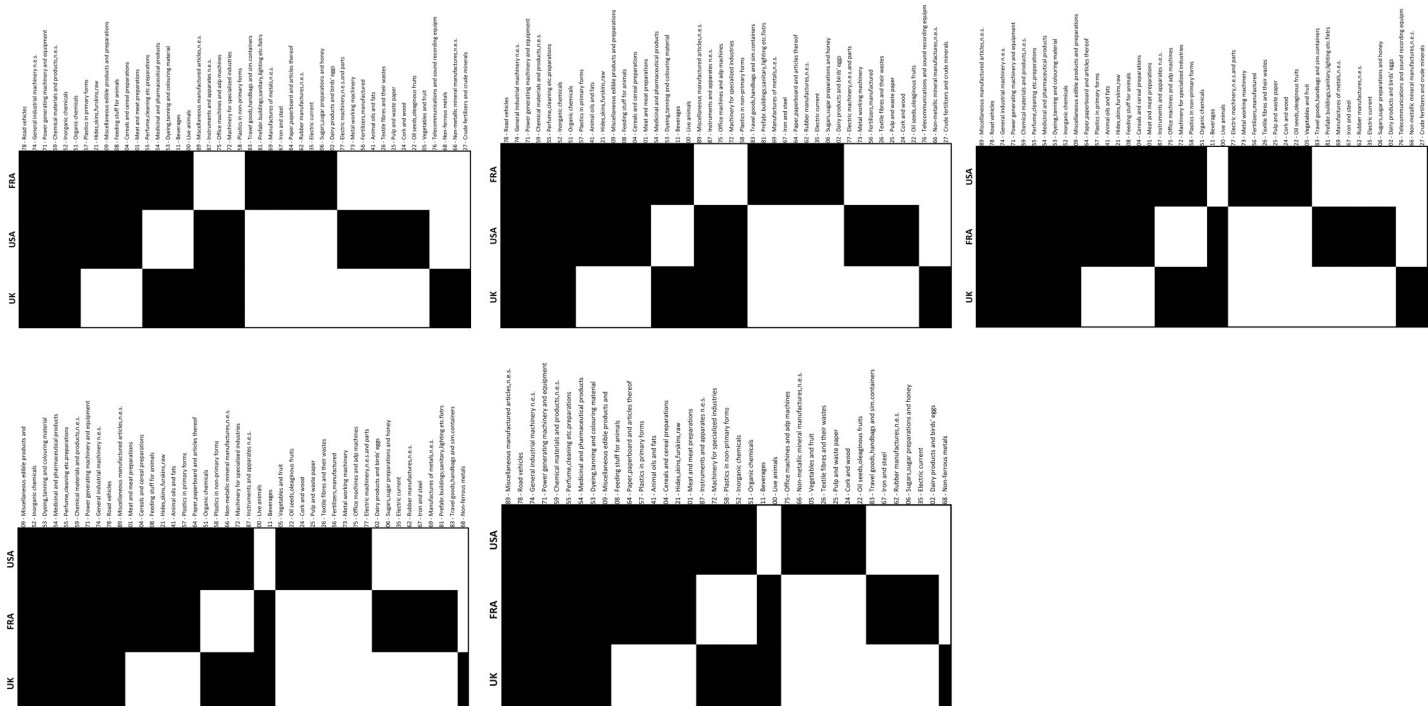

**Fig 6. Evolution of packed matrices. a.** Group 1: FRA, the UK and the USA, 1992–1996. **b.** Group 1: FRA, the UK and the USA, 1997–2001. **c.** Group 1: FRA, the UK and the USA, 2002–2006. **d.** Group 1: FRA, the UK and the USA, 2007–2011. **e.** Group 1: FRA, the UK and the USA, 2012–2016.

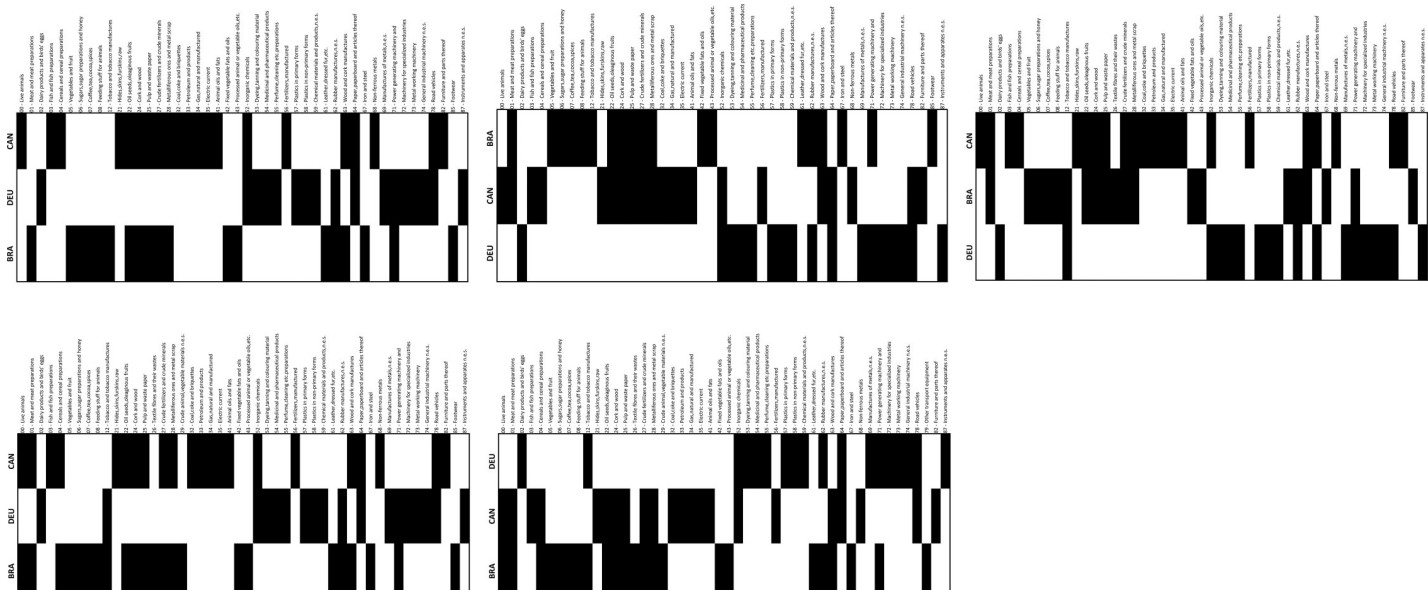

**Fig 7. Evolution of unpacked matrices. a.** Group 2: BRA, CAN and DEU, 1992–1996. **b.** Group 2: BRA, CAN and DEU, 1997–2001. **c.** Group 2: BRA, CAN and DEU, 2002–2006. **d.** Group 2: BRA, CAN and DEU, 2007–2011. **e.** Group 2: BRA, CAN and DEU, 2012–2016.

Each matrix is compared to row-column proportional (PP) null models, as this is the most stringent and widely used among scientist to assess nestedness significance [11]. The evolution of the nestedness measures of both groups and results of the PP null models for each metric is illustrated in Fig 9A–9C.

Results evidence that the country-product networks are nested. Both groups depict higher nestedness across all metrics when compared with the PP null model. Likewise, developed

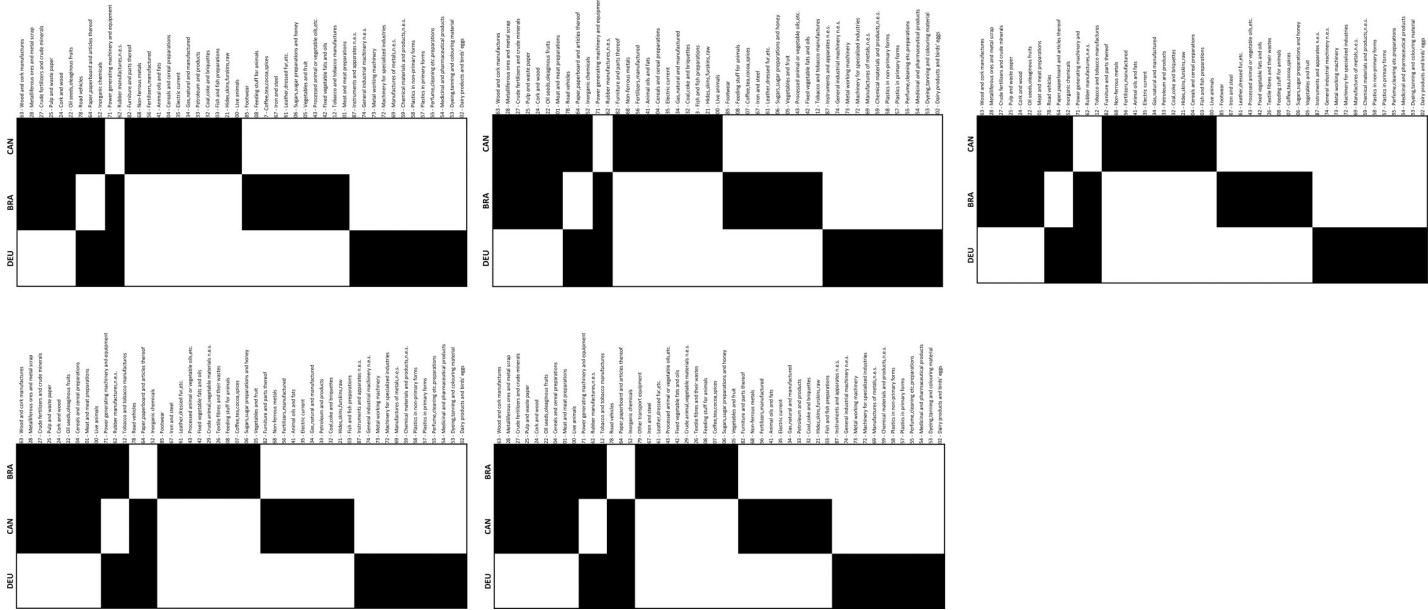

**Fig 8. Evolution of packed matrices. a.** Group 2: BRA, CAN and DEU, 1992–1996. **b.** Group 2: BRA, CAN and DEU, 1997–2001. **c.** Group 2: BRA, CAN and DEU, 2002–2006. **d.** Group 2: BRA, CAN and DEU, 2007–2011. **e.** Group 2: BRA, CAN and DEU, 2012–2016.

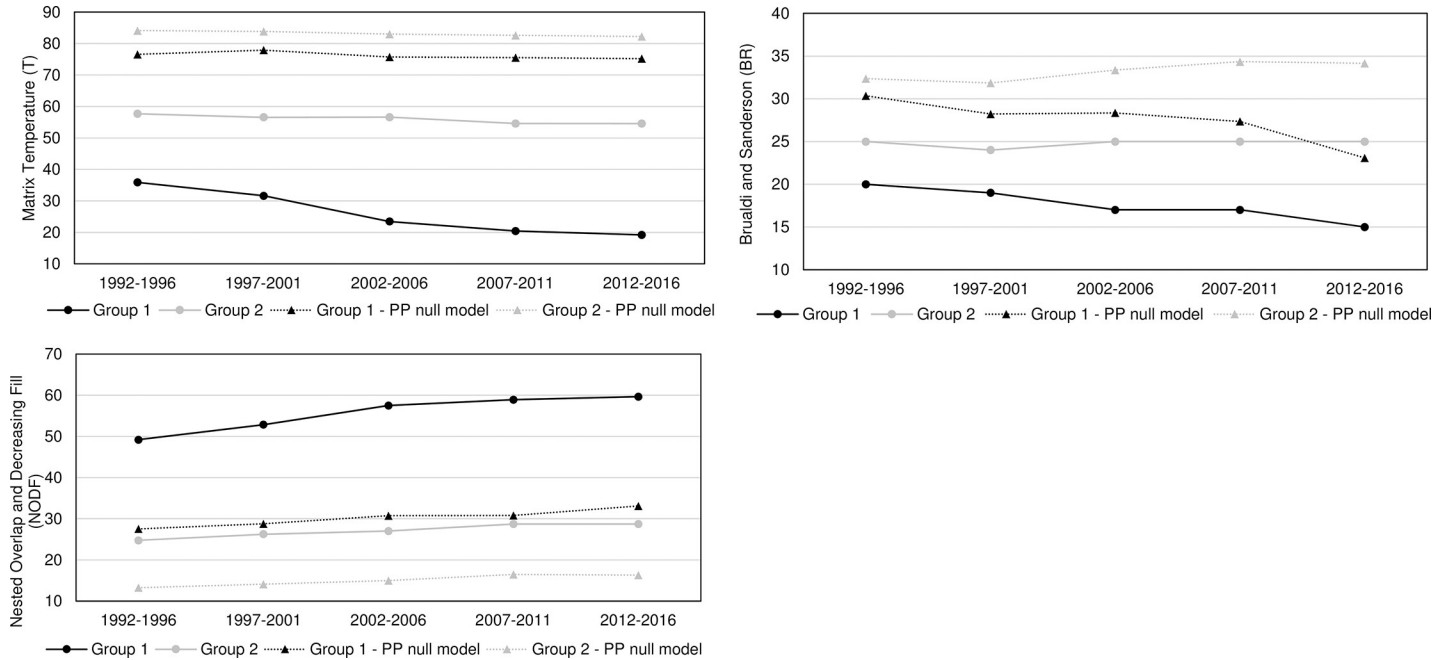

**Fig 9. Nestedness measurement and validation with PP null models. a.** Matrix Temperature. **b.** Brualdi and Sanderson. **c.** Nested overlap and decrease fill.

ecosystems, group 1, have greater nestedness than the less developed ones, group 2. Thus, it has been demonstrated that nestedness increases accordingly to the aerospace ecosystems evolution. A deeper analysis is offered in the following sections.

**Microscopic level.** *Degree centrality* is a metric used to compute the number of direct connections to a node. This measure is used to identify the most popular RCA products within the ecosystems; where a higher degree reflects that more countries have an RCA>1 on a specific product [52].

Figs 10 and 11 give information on the degree centrality of all the products for group 1 and group 2, respectively. In these graphs, degree centrality is represented by colours: the darker the colour, the higher the degree centrality. Here, the value of degree centrality is directly related to the number of countries that have developed an RCA>1 on that product. For instance, a degree centrality of 3 (the darkest blue) means that three countries have an RCA>1 on that product; a degree centrality of 0 (lightest blue) evidences that no country has developed an RCA>1 on that product. An interesting fact is that group 2 does not have any product with a degree centrality of 3, which means that the three countries do not have a common product in which all of them have developed an RCA>1.

## Interpretation of results

### Revealed comparative advantage

From 1992 to 2016, a number of events influenced the economy worldwide and consequently, the aerospace ecosystem. Some examples include the early 1990s recession in the European Union and the USA, 'black Wednesday' in the UK in 1992, Asian financial crisis in 1997, Russian financial crisis in 1998, early 2000s recession, 9/11 terrorist attacks in the USA in 2001, the global financial crisis of 2007–2008, debts crisis in Greece, Ireland, Italy, Portugal and Spain starting in 2009, and other particular country-level events [53,54]. Some of these events may have caused RCA fluctuations observed in Fig 1C.

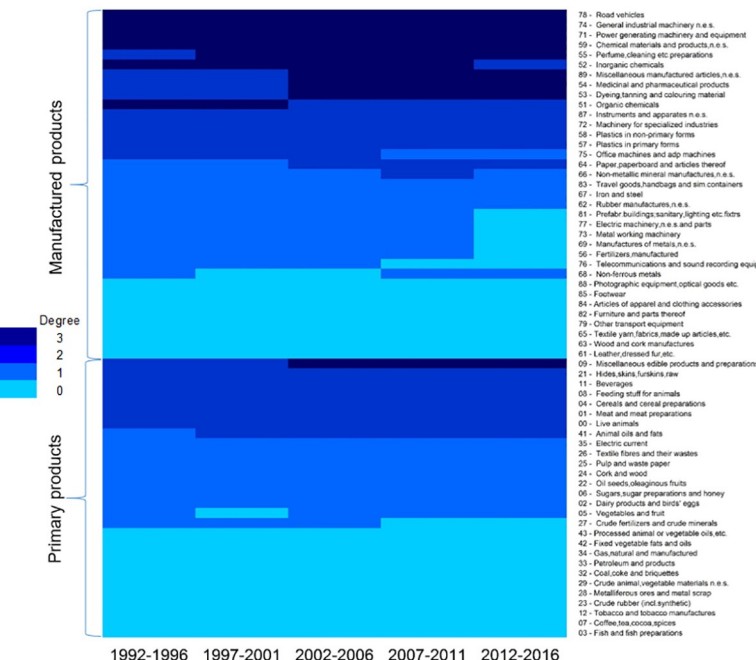

**Fig 10. Degree centrality for group 1.**

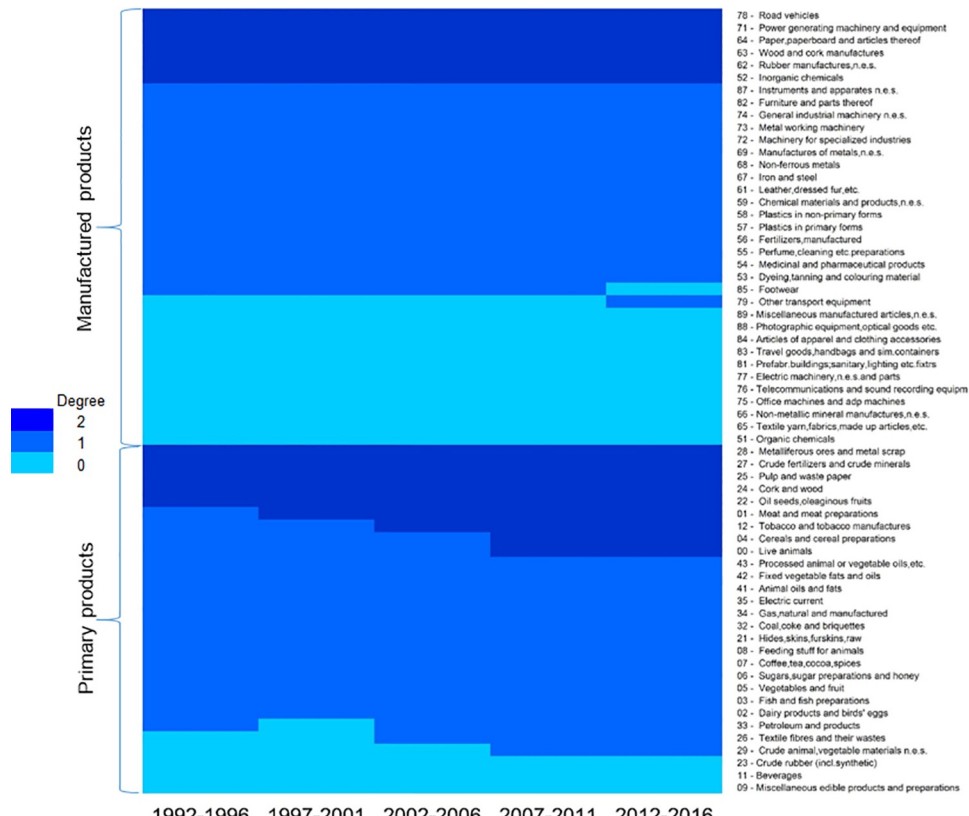

**Fig 11. Degree centrality for group 2.**

Overall, the USA exports most Aerospace products in the world, followed by FRA, DEU, the UK and CAN. However, countries with the highest value of Aerospace exports do not necessarily have a superior RCA. Group 1 depicted a consistent RCA>1 during the period of study. The RCA average on aerospace products for this group grew steadily during the first three periods of study, with a value of 2.8 in 1992 up to 4.3 in 2012. Since 2013, this value slightly decreased down to 3.9 in 2016. Group 2 has demonstrated a lower RCA than group 1; starting to increase from 1998. Also, this group of countries achieved their peak at the end of 2000, mainly driven by the development of Brazil's aerospace ecosystem.

At the country level, the RCA on aerospace products presents two main oscillations. The first one was experienced by BRA starting in 1999. BRA officially started its aerospace industry in 1941, when they created the governmental agency named Ministry of Aeronautics (MAER). A few years later, in 1945, they formed the Technical Centre of Aeronautics (CTA), aiming to promote the development of this sector. In 1950, they opened their first engineering school focused on aeronautics, named the Institute for Aerospace Technology (ITA). In 1969, Brazil's government founded EMBRAER, the Brazilian aerospace manufacturer, and in 1994 this company was denationalised. After privatisation, in 1999, BRA started to develop an RCA>1 on aerospace products. During the same year, BRA experienced a currency devaluation against the US Dollar, just a year after the Russian financial crisis. Both, the EMBRAER´s privatisation and devaluation of the Brazilian real, could have been the enablers behind achieving the aerospace industry´s peak in 2000, followed by an abrupt decrease.

The second main fluctuation is observed in 2010 when the French aerospace ecosystem grew. FRA is mainly an importer of components and equipment, and a final assembler and exporter of aeroplanes and helicopters, representing almost 65% of their aerospace exports. In recent years, the aerospace industry in this country has been one of the most important [5]. The importance of this sector in its national economy is higher than it is for other key players. For instance, in 2015, 3.5% of its GDP was due to exports of aerospace products, whereas in countries such as the USA, the UK and CAN it represented only around 0.7% [5]. Since the early 2000s, the French aerospace ecosystem gradually rose thanks to the sharp growth of air traffic, particularly from the Asia-Pacific region. Its RCA peaked in 2010–2012, just after the global financial crisis of 2007–2008, and after the USA slowed down after steady growth since 2000. This could have been driven predominantly by the increase in passenger demand, from the Asia-Pacific region. Singularly, 2010 is considered to be a year when the air traffic demand experienced a breakthrough [5]. During this year, the numbers of passengers carried increased by nearly 17% from the previous year (from 2.25 in 2009 to 2.628 billion passengers in 2010) [55]. Starting in 2013, the RCA steeply dropped mainly because manufacturers experienced a lack of sufficient production capacity and a sharp fall in demand lead from oil-producing countries [5].

## Networks analysis: Macroscopic level

The analysis of the country-product networks developed in this study has helped us to identify patterns in the evolution of developed aerospace ecosystems. The patterns that have characterised the ecosystems' development at a macroscopic level are presented next.

Network density helps us to evidence that networks increase their cohesiveness as their aerospace ecosystem develops. The increase in cohesiveness is driven by an increase in the number of actual versus potential connections. This means that countries tend to have fewer isolated nodes and more shared products with other countries. For instance, products that are connected only to one country for group 1 decreased from 22 in the first period, to 14 in the last period. Concerning the group of less developed countries, the number of nodes unique to

a single country is considerably higher than more developed countries. This group started with 35 nodes and decreased down to 33 during the last period.

In regards to network centralisation, this metric evidences that networks tend to centralise power in fewer nodes by creating larger clusters with shared products in the networks. For instance, for group 1, the number of products shared between the three countries rose from 6 in the first period up to 9 in the last period.

## Nestedness analysis

Inspired by studies in other fields such as biological ecosystems, in this work, we searched for nested patterns in the bipartite country-products networks. Measuring their nestedness using three widely applied metrics and comparing them with randomly generated networks, it was shown that the networks developed in this work are nested. More importantly, it has been demonstrated that more developed ecosystems present a higher level of nestedness and that it increases in tandem with an RCA>1 in the aerospace ecosystems.

The packed matrices in Figs 6A–6E and 8A–8E show a typical behaviour of how nestedness patterns are exposed after reordering the original matrices: increasing presences of country-products in the top left corner of each graph. Patterns reveal that countries tend to increase their diversification by developing an RCA>1 on more products rather than specialising only on one. Moreover, it is revealed that although countries develop an advantage on unique products, they increase competition with each other as they incline to develop an RCA>1 on a specific group of products. Nodes tend to form larger clusters in the centre of the networks, meaning that as the countries' aerospace ecosystem develops, the number of shared products with other countries tends to increase. Thus, countries lean towards having an RCA>1 within the same group of products, evidencing that their ecosystems also tend to become more similar.

Nestedness analysis in this research has also contributed to confirm that mutualistic interaction patterns originally found in species-species networks are also found across networks of different nature. Nestedness patterns found in the country-product networks developed in this research are particularly aligned with the hypothesis that most common relations occur between generalist-generalist, and that specialist are mainly related to generalist [28,29]. The latter hypothesis, specialist products produced mainly by generalist countries, is observed through the evolution of nestedness across different periods as it increases over time, and in particular, more pronounced on the packed matrices of group 1 (Fig 6A–6E). For instance, the UK aerospace ecosystem as a specialist country, positioned at the bottom of Fig 6A–6E matrices, tends to reduce over time the number of specialist products and increase the generalist products. A similar scenario is depicted for the country situated in between the three countries (the USA during the first two periods and FRA during the last three periods), where the amount of generalist products tends to increase and the specialist products to reduce over time. A bit less evident but still identifiable, this hypothesis is also observed in group 2 through the evolution of Fig 8A–8E matrices. This is expected as nestedness of group 2 is lower and presents a smaller increase over time than group 1, as evidenced on results shown on Fig 9A–9C. Previous findings are also aligned with studies developed on networks from other industrial sectors, such as inter-organisational networks and networks from the automotive sector. For instance, patterns found in manufacturer-contractor interaction networks by [35] in which they found similar patterns than the mutualistic interaction patterns between species-species networks. Patterns found in automotive supply chain networks by [36] in which they showed that generalist companies are the only ones producing specialist products and that specialists companies compete practically utterly in the generalist products market. It is also

aligned with the study presented in [37], where they analysed the supplier-product distribution and supplier-manufacturer relations in the global automotive industry. They claim that specialist suppliers produce proper subsets of what generalist suppliers produce, and that specialist products are only produced by generalist suppliers.

## Networks analysis: Microscopic analysis

The microscopic analysis also helped us to identify the specific products that have correlated with the growth of the aerospace ecosystems over the last 25 years. Figs 10 and 11 show the evolution of the product competition within groups. In these graphs, the degree centrality is directly linked with the number of countries that have developed an RCA>1 on that product. For group 1, the amount of products with a degree centrality higher than 0 is higher in manufactured products than primary products. Here, manufactured products represent 61% of the products. That is not the case for group 2, as these countries have a more balanced product portfolio with 51% represented by primary products. Such difference between primary and manufactured products diversification in the countries of each group could be because primary products are more dependent on the geographical location, climate and biodiversity of each country rather than choice or strategy.

Similar to the finding from the nestedness analysis, the microscopic analysis helps to reinforce the hypothesis that countries with developed aerospace ecosystems tend to increase their diversification in tandem with their aerospace evolution, by developing an RCA>1 on more products rather than specialising only on one. This means that the number of products with an RCA>1 per country increases simultaneously with an increase in the RCA on aerospace products. For instance, group 1 increased from having a total of 72 links country-product on the 1992–1996 period, up to 76 on the 2007–2011 period. At the same time, the RCA on Aerospace products for this group increased from an average of 2.6 in 1992, up to 4.2 in 2011. In contrast, the number of country-products links and RCA average on aerospace products decreased simultaneously throughout the last period. During the 2012–2016 period, the number of country-product links decreased down to 69, accompanied by a decrease in 2016 equivalent to 0.3 points on the RCA on Aerospace products, compared to 2011. For group 2, the number of country-products links increased from 57 in the first period, up to 63 in the 2007–2011 period, while the RCA average on aerospace products increased from an average of 0.7 in 1992, up to an average of 1.4 in 2011. For this group, both figures remained constant during the last period of study. Previous findings are aligned with [18,34], in which they claim that developed countries are highly diversified. The principal added value of our analysis is the identification of which particular industries have contributed the most with aerospace ecosystems development.

As can be seen in Fig 1C, the RCA on aerospace products for group 1 depicts an upward trend until the third period, and experience a slight decrease during the last period. A similar pattern is found in the products with the highest degree centrality, as shown in Fig 10. The number of shared products by the three countries started with six during the first period, increased to seven during the second period, to ten during the third period, remained steady during the fourth period and finally decreased to nine during the fourth period. Apart from product '09 –Miscellaneous edible products and preparations' which increased its degree centrality from two during the second period to three during the third period, all the other products that increased the degree centrality are manufactured products. For group 1, *road vehicles*, *general industrial machinery*, *power generating machinery and equipment*, and *chemical materials and products* have been the products that the three countries have competed during the period of study.

In contrast, group 2 does not have any product that is common for the three countries, and most of the products are unique to a single country. Countries of this group maintained unchanging the degree centrality distribution on manufactured goods: the same six products with the highest degree centrality during all the period of study. In regards to primary products, the number of products with the highest degree centrality increased from 11 to 12, then to 13, and finally to 15 during the last two periods. The products with the highest degree centrality among all the period of study for this group are *road vehicles*, *power generating machinery and equipment*, *paper, paperboard and articles thereof*, *wood and cork manufactures*, *rubber manufactures*, *inorganic chemicals*, *metalliferous ores and metal scrap*, *crude fertilizers and crude minerals*, *pulp and waste paper*, *cork and wood*, and *oil seeds, oleaginous fruits*. It is relevant to highlight that *road vehicles* and *power generating machinery and equipment* are the only products that are part of the shared portfolio of group 1.

Regarding the products that have been correlated with the aerospace sector, manufactured products (91 for group 1 and 77 for group 2 'correlation links') depicted a stronger correlation than primary products (52 for group 1 and 50 for group 2 'correlation links'). For group 1, *road vehicles* and *medicinal and pharmaceutical products* have been the most correlated with the aerospace evolution. For group 2, excluding the first period as the group did not have an RCA>1 in aerospace products, *road vehicles*, *power generating machinery and equipment*, *wood and cork manufactures*, *rubber manufactures*, *paper, paperboard and articles thereof*, and meat and meat preparations have been the most correlated with the aerospace evolution. *Road vehicles* are the only correlated products in common for both groups.

## Conclusions

Inspired by studies that have developed economic theories and analysed the behaviour of industrial ecosystems by taking a network science approach, in this work we used historical trade data and network theory to find patterns that have characterised the evolution of developed aerospace manufacturing countries ecosystems. First, we used export data over 25 years and computed the RCA on aerospace products to identify the countries subject to study. Two groups of countries were created: group 1 –France, the USA and the UK, group 2 –Brazil, Canada and Germany. The first group of countries maintained an RCA>1 on aerospace products among the 25 years. The second group started with an RCA>1 approximately in 1998 and maintained it until the final year of study. Results evidenced that countries with the highest value of aerospace exports do not necessarily have a superior advantage when compared with other countries. Subsequently, we developed bipartite country-product networks and found a number of patterns that have distinguished the evolution. Motivated by studies in ecological networks, we used nestedness to find patterns in the distribution and evolution of exported products across ecosystems. The analysis presented in this research contributes to confirming that mutualistic interaction patterns originally found in species-species networks are also found across networks of different nature. Networks developed in this research are aligned with the claim that most popular interactions occur between generalist-generalist, and that specialist are mainly related to a generalist.

It was revealed that developed aerospace ecosystems lean towards increasing the number of products with an RCA>1 together with their aerospace evolution, which means that the number of products with an RCA>1 per country increases simultaneously with an increase in the RCA on aerospace products. This finding is aligned with previous studies from [18,34], in which they claim that developed countries are highly diversified; it is also contrary to the hypothesis of [16] that developing countries tend to have a high product diversification.

Another finding is that developed aerospace ecosystems also tend to become more similar. Although countries increase their diversification and develop an advantage on unique

products, they also increase the competition with each other as they incline to develop an RCA>1 on a specific group of products. This means that as the countries' aerospace ecosystem develops, the amount of RCA>1 products shared with other countries tends to increase. Thus, countries lean towards having an RCA>1 in the same group of products.

The network analysis at a microscopic level was used to reveal the specific products that have nourished the growth of the aerospace ecosystems over the last 25 years. For the group of more developed ecosystems (the USA, the UK and FRA), *road vehicles*, *general industrial machinery*, *power generating machinery and equipment*, and *chemical materials* have been the most popular products throughout all the period of study. The other group of countries (BRA, CAN and DEU), with less developed aerospace ecosystems, does not have any product that is popular for the three countries, and most of the products are unique to a single country. The only products that all the countries of group 1 and group 2 (excluding Brazil) have developed an RCA>1 are *road vehicles* and *power generating machinery and equipment'*. Thus, the automotive and the power generating machinery and equipment products are the most popular that developed aerospace ecosystems tend to compete.

Concerning the products that have been correlated with the aerospace ecosystem, generally manufactured products have depicted a stronger correlation than primary products. Moreover, results revealed that group 2 has more positively correlated products than group 1. It is relevant to highlight that all over all the period of study, the automotive sector has been the most popular on having a positive correlation with the aerospace ecosystem.

## Limitations and further research

This research can be further complemented with the utilisation of identification methods based on input-output methods to identify the relationship between products. In this research, network science is selected because it is an emerging technique that has already contributed to the field by providing measures that are agnostic to preconceived hypotheses on reasons of similarity between products.

Indeed, highly cited works of Hidalgo and Hausmann make a similar observation [17,19,42,49], suggesting that mainstream economics has thus far followed two main approaches to explain a country's pattern of specialisation: first of these is a relative proportion between productive factors (which suggests that poor countries specialize in goods that are produced using unskilled labour and land while richer countries specialise in goods requiring infrastructure, institutions, human and physical capital) and second of these approaches emphasise technological differences.

Instead, they observed that these methods fail to capture complexities such as cold storage systems used to produce fresh fruit giving an existing infrastructure for other products that require similar infrastructure. They explicitly assert that input-output methods also have preconceived hypotheses on similarity: they measure the relatedness of products based on input/outputs involved in a product's value chain (e.g. cotton, yarn, cloth, garments). While there are certainly many cases where this holds true; using only input-output methods would confine us to the value chain hypotheses on relatedness, and as such, we might lose sight of other complexities in the development of an aerospace system. Our anecdotal discussions with aerospace producers in fact confirm this, suggesting for example that existing car manufacturing industry is a beneficial factor to aerospace growth because of synergy in additive manufacturing and composite production.

This is the primary reason we choose network science, and in particular, Revealed Comparative Advantage measure, based on the idea that if two products are related, for whatever reasons then they will tend to be produced together. It is also worth noting that our approach then enables us to do a post hoc analysis and explore why products might be related.

In addition, it is suggested that the methodology may be applied to the study of other industrial ecosystems. For instance, if a country wants to foster its pharmaceutical and medicinal ecosystem, it might need first to analyse the patterns and key enablers found in developed ecosystems so they can emulate their evolution.

A further limitation of this study is data availability. The two-digit SITC commodities classification was the most complete database available at the moment when this research was elaborated. A more specific commodities' classification may significantly contribute to propose more specific recommendations.

A detailed analysis of the reasons behind the aerospace ecosystems evolution and of the causality in the relationships is also a limitation. While the scope of this research has been limited to the identification of patterns across the product space evolution of developed aerospace ecosystems, further research aiming at the identification of reasons behind such patterns is necessary.

## Author Contributions

**Conceptualization:** Alexandra Brintrup, Konstantinos Salonitis.

**Formal analysis:** Luna A. Jose, Jr.

**Methodology:** Luna A. Jose, Jr.

**Project administration:** Luna A. Jose, Jr.

**Resources:** Alexandra Brintrup.

**Supervision:** Alexandra Brintrup, Konstantinos Salonitis.

**Validation:** Alexandra Brintrup, Konstantinos Salonitis.

**Writing – original draft:** Luna A. Jose, Jr., Alexandra Brintrup, Konstantinos Salonitis.

**Writing – review & editing:** Luna A. Jose, Jr., Alexandra Brintrup, Konstantinos Salonitis.

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
