## [Decision Letter · Decision Letter 0]

26 Mar 2020

PONE-D-19-35348

Analysis of the evolution of developed aerospace manufacturing countries’ ecosystems: identification of patterns and similarities using bipartite country – product networks based on trade data over 25 years

PLOS ONE

Dear Mr. Luna A.,

Thank you for submitting your manuscript to PLOS ONE. After careful consideration, we feel that it has merit but does not fully meet PLOS ONE’s publication criteria as it currently stands. Therefore, we invite you to submit a revised version of the manuscript that addresses the points raised during the review process.

I recommend that it should be revised taking into account the changes requested by Reviewers. I would like to give you the last chance to revise your manuscript. o speed the review process, the manuscript will only be reviewed by the Academic Editor in the next round.

We would appreciate receiving your revised manuscript by May 10 2020 11:59PM. To enhance the reproducibility of your results, we recommend that if applicable you deposit your laboratory protocols in protocols.io, where a protocol can be assigned its own identifier (DOI) such that it can be cited independently in the future. For instructions see: http://journals.plos.org/plosone/s/submission-guidelines#loc-laboratory-protocols

We look forward to receiving your revised manuscript.

Kind regards,

Baogui Xin, Ph.D.

Academic Editor

PLOS ONE

Journal Requirements:

2. Please upload a copy of Figures 12 - 25, to which you refer in your text. If the figures are no longer to be included as part of the submission please remove all reference to them within the text.

Reviewers' comments:

Reviewer's Responses to Questions

**Comments to the Author**

1. Is the manuscript technically sound, and do the data support the conclusions?

Reviewer #1: Yes

Reviewer #2: Yes

2. Has the statistical analysis been performed appropriately and rigorously? 

Reviewer #1: Yes

Reviewer #2: No

3. Have the authors made all data underlying the findings in their manuscript fully available?

Reviewer #1: Yes

Reviewer #2: Yes

4. Is the manuscript presented in an intelligible fashion and written in standard English?

Reviewer #1: Yes

Reviewer #2: Yes

5. Review Comments to the Author

Reviewer #1: This paper uses the trade data of 25 years to study the evolution of developed aerospace manufacturing countries’ ecosystems. It is a very interesting problem. The development of aviation field has a certain access threshold, which has a high demand for a country's scientific research strength. Based on the network theory, this paper finds out the characteristic pattern of ecosystem evolution in developed aerospace manufacturing countries. The internal logic of the evolution of space ecosystem can be deeply analyzed by using the network theory. The research in this paper is worthy of affirmation.

It needs the author to modify the abstract of the article, so that the abstract part can reflect the methods used in this paper and the conclusions obtained. From the current abstract, the research conclusions of this paper are relatively vague and can be further improved.

I suggest that the journal accept this article.

Reviewer #2: The current manuscript uses network science-based methodologies to analyse the evolution of the aerospace sector. In this study, the authors develop bipartite country – product networks based on trade data over 25 years, aiming to identify patterns and similarities in the evolution of developed aerospace manufacturing countries ecosystems.

I like this work, and I think after some revision it will be suitable for publication in the Plos One.

Therefore, I would ask the authors to consider the following comments, which I believe will improve and clarify some parts of their manuscript.

1. This paper only selects France, Britain, the United States, Brazil, Canada and Germany. What is the basis of the selection? The results of this paper may be biased. The author should explain the process of selecting these countries so as to make the results more convincing.

2. This paper selects cointegration method to identify the relationship between products, but as far as I know, most of the existing researches use product space and input-output methods to identify the relationship between products. These methods are more reliable than cointegration relations. The author can use more identification methods as robustness test.

3. The article only reveals the current situation of the aerospace systems, but does not explain the reasons. The identification of causality is not rigorous, for example, in line 483, "it has been determined that nested increases according to the aerospace systems evolution", or it may be that the increase of nested increases in turn promotes the aerospace systems evolution

6. PLOS authors have the option to publish the peer review history of their article (what does this mean?). If published, this will include your full peer review and any attached files.

Reviewer #1: No

Reviewer #2: No

---

## [Author Response · Author response to Decision Letter 0]

2 Apr 2020

Associate Editor Comments to the Author:

Thank you for submitting your manuscript to PLOS ONE. After careful consideration, we feel that it has merit but does not fully meet PLOS ONE’s publication criteria as it currently stands. Therefore, we invite you to submit a revised version of the manuscript that addresses the points raised during the review process.

I recommend that it should be revised taking into account the changes requested by Reviewers. I would like to give you the last chance to revise your manuscript. To speed the review process, the manuscript will only be reviewed by the Academic Editor in the next round.

Response to Associate Editor:

We thank the editor, associate editor and all reviewers for their insightful comments and support. The short duration of the review process, and clear guidelines for the revision were very much appreciated. 

We have considered your suggestions and your feedback has helped to further improve the paper. All change requests were implemented.

Below is a detailed list of comments raised by each reviewer followed by our responses. 

In this report, we summarise our responses to reviewer comments. Changes in the manuscript have been highlighted in yellow. 

Reviewer 1

Overall remarks: This paper uses the trade data of 25 years to study the evolution of developed aerospace manufacturing countries’ ecosystems. It is a very interesting problem. The development of aviation field has a certain access threshold, which has a high demand for a country's scientific research strength. Based on the network theory, this paper finds out the characteristic pattern of ecosystem evolution in developed aerospace manufacturing countries. The internal logic of the evolution of space ecosystem can be deeply analyzed by using the network theory. The research in this paper is worthy of affirmation.

I suggest that the journal accept this article.

Comment 1.1: It needs the author to modify the abstract of the article, so that the abstract part can reflect the methods used in this paper and the conclusions obtained. From the current abstract, the research conclusions of this paper are relatively vague and can be further improved.

Response 1.1: We thank the reviewer for this suggestion. We have now revised the abstract and incorporated the main conclusions from the study, as follows: 

“Countries also tend to become more nested in their aerospace product space as they start developing a higher RCA. It is revealed that although countries develop an advantage on unique products, they also tend to increase competition with each other. Further analysis shows that, manufactured products have a stronger correlation to an aerospace ecosystem, than primary products; and in particular, the automotive sector shows the highest correlation with positive aerospace sector evolution. Competition between countries with well-developed aerospace ecosystems tends to centre on automotive parts, general industrial machinery, power generating machinery and equipment, and chemical materials and products.”

Reviewer 2 

Overall remarks: The current manuscript uses network science-based methodologies to analyse the evolution of the aerospace sector. In this study, the authors develop bipartite country – product networks based on trade data over 25 years, aiming to identify patterns and similarities in the evolution of developed aerospace manufacturing countries ecosystems.

I like this work, and I think after some revision it will be suitable for publication in the Plos One.

Therefore, I would ask the authors to consider the following comments, which I believe will improve and clarify some parts of their manuscript.

Comment 2.1: This paper only selects France, Britain, the United States, Brazil, Canada and Germany. What is the basis of the selection? The results of this paper may be biased. The author should explain the process of selecting these countries so as to make the results more convincing.

Response 2.1: 

Countries were selected based on their RCA values over the study period. We first conducted an RCA analysis to determine which countries. Two groups of countries are needed for our analysis: one group have consistently been among the top on aerospace products, and another group of countries that has improved their exports capability on aerospace products by moving from RCA<1, to RCA>1 in the period examined (explained in page 17). By comparing the two groups we are then able to cross-identify product export patterns that nurture a developed aerospace ecosystem. 

Comment 2.2: This paper selects cointegration method to identify the relationship between products, but as far as I know, most of the existing researches use product space and input-output methods to identify the relationship between products. These methods are more reliable than cointegration relations. The author can use more identification methods as robustness test.

Response 2.2: 

It is true that much work to analyse product space is based on input-output methods. On the other hand, network science is an emerging technique that has already contributed to the field by providing measures that are agnostic to preconceived hypotheses on reasons of similarity between products, which is the reason we chose it.

Indeed, highly cited works of Hidalgo and Hausmann make a similar observation (Hidalgo et al. 2007; Hidalgo and Hausmann 2009; Hausmann and Hidalgo 2010; Bahar, Hausmann, and Hidalgo 2014), suggesting that mainstream economics has thus far followed two main approaches to explain a country’s pattern of specialisation: first of these is relative proportion between productive factors (which suggests that poor countries specialize in goods relatively intensive in unskilled labour and land while richer countries specialise in goods requiring infrastructure, institutions, human and physical capital) and second of these approaches, emphasise technological differences. 

Instead, they observed that these methods fail to capture complexities such as cold storage systems used to produce fresh fruit giving an existing infrastructure for other products that require similar infrastructure. They explicitly assert that input-output methods also have preconceived hypotheses on similarity: they measure relatedness of products based on input / outputs involved in a product’s value chain (e.g. cotton, yarn, cloth, garments). While there are certainly many cases where this holds true; using only input-output methods would confine us to the value chain hypotheses on relatedness and as such, we might lose sight of other complexities in the development of an aerospace system. Our anecdotal discussions with aerospace producers in fact confirmed this, suggesting for example that existing car manufacturing industry is a beneficial factor to aerospace growth because of a synergy in additive and composite production. 

This is primary reason we choose network science, and in particular, Revealed Comparative Advantage measure, based on the idea that if two goods are related, for whatever reasons (because they require similar institutions, infrastructure, physical factors, technology, or some combination thereof) then they will tend to be produced together. It is also worth noting that our approach then enables us to do a posthoc analysis and explore why products might be related. 

We have now added a “Limitations and Further Research” Section to the article, which describes our reasoning behind the selection of this methodology and emphasises the need to conduct further, complimentary macro- economic analysis approaches. 

Comment 2.3: The article only reveals the current situation of the aerospace systems, but does not explain the reasons. The identification of causality is not rigorous, for example, in line 483, "it has been determined that nested increases according to the aerospace systems evolution", or it may be that the increase of nested increases in turn promotes the aerospace systems evolution.

Response 2.3:

A detailed analysis of the reasons behind the aerospace ecosystems evolution and of the causality in the relationships is also a limitation. While the scope of this research has been limited to the identification of patterns across the product space evolution of developed aerospace ecosystems, further research aiming at the identification of reasons behind such patterns is necessary. 

Concerning the rigorousness in the identification of causality relationships, we also believe that additional analyses can also be elaborated to strengthen the findings and for the identification of the direction of relationships. For instance, it has been identified that developed aerospace ecosystems tend to become more nested in their aerospace product space as they start developing a higher RCA. It is worth further analysis to identify the direction on the causality of such relationship. 

We have now added a “Limitations and Further Research” Section to the article, which mentions these limitations and highlights the need for further research in causality analysis.

---

## [Editor Report · Decision Letter 1]

6 Apr 2020

Analysing the evolution of aerospace ecosystem development

PONE-D-19-35348R1

Dear Dr. Luna A.,

We are pleased to inform you that your manuscript has been judged scientifically suitable for publication and will be formally accepted for publication once it complies with all outstanding technical requirements.

With kind regards,

Baogui Xin, Ph.D.

Academic Editor

PLOS ONE
---

## [Editor Report · Acceptance letter]

10 Apr 2020

PONE-D-19-35348R1 

Analysing the evolution of aerospace ecosystem development 

Dear Dr. Luna A.:

I am pleased to inform you that your manuscript has been deemed suitable for publication in PLOS ONE. Congratulations! Your manuscript is now with our production department. 

With kind regards,

on behalf of

Prof. Baogui Xin 

Academic Editor

PLOS ONE